# Connecting moss lipid droplets to patchoulol biosynthesis

**Anantha Peramuna**[1], **Hansol Bae**[1,2], **Carmen Quiñonero López**[1], **Arvid Fromberg**[3], **Bent Petersen** [id][4,5], **Henrik Toft Simonsen** [id][1,2]*

**1** Department of Biotechnology and Biomedicine, Technical University of Denmark, Lyngby, Denmark, **2** Mosspiration Biotech, Hørsholm, Denmark, **3** National Food Institute, Technical University of Denmark, Lyngby, Denmark, **4** The GLOBE Institute, University of Copenhagen, Copenhagen, Denmark, **5** Centre of Excellence for Omics-Driven Computational Biodiscovery, AIMST University, Kedah, Malaysia

* hets@dtu.dk

## Abstract

Plant-derived terpenoids are extensively used in perfume, food, cosmetic and pharmaceutical industries, and several attempts are being made to produce terpenes in heterologous hosts. Native hosts have evolved to accumulate large quantities of terpenes in specialized cells. However, heterologous cells lack the capacity needed to produce and store high amounts of non-native terpenes, leading to reduced growth and loss of volatile terpenes by evaporation. Here, we describe how to direct the sesquiterpene patchoulol production into cytoplasmic lipid droplets (LDs) in *Physcomitrium patens* (syn. *Physcomitrella patens*), by attaching patchoulol synthase (PTS) to proteins linked to plant LD biogenesis. Three different LD-proteins: Oleosin (*Pp*OLE1), Lipid Droplet Associated Protein (*At*LDAP1) and Seipin (*Pp*Seipin325) were tested as anchors. Ectopic expression of PTS increased the number and size of LDs, implying an unknown mechanism between heterologous terpene production and LD biogenesis. The expression of PTS physically linked to Seipin increased the LD size and the retention of patchoulol in the cell. Overall, the expression of PTS was lower in the anchored mutants than in the control, but when normalized to the expression the production of patchoulol was higher in the seipin-linked mutants.

## Introduction

Plant-derived specialized metabolites such as terpenoids are exploited for their value in fragrances, flavors, and pharmaceutics industries [1]. The natural productions of several of these fragrant terpenoids are unsustainable due to agricultural and environmental factors, which is reflected in excessive use of water and the wholesale price. Patchouli oil, where the major component is patchoulol from *Pogostemon cablin* has a fluctuating market price of 30–200 US dollars/Kg. Production of terpenoids in a heterologous expression system (yeast, algae or moss) has had some successes [2–6]. However, high yields and efficient extraction methods comparable to the native plants are still a major challenge. Native plants have evolved to produce and store volatile terpenoids in specialized cells [7, 8]. Heterologous expression systems lack the cellular specialty needed to contain and produce high amounts of these compounds without

**Data Availability Statement:** All relevant data are within the manuscript and its Supporting Information files.

**Funding:** AP, HB, and HTS were supported by The Danish Council for Independent Research (#4005-

00158B). CQ was supported by funding from the People Program (Marie Curie Actions) of the European 517 Union's Seventh Framework Program FP7/2007-2013/ under REA grant agreement n° [607011]. The funders had no role in study design, data collection and analysis, decision to publish, or preparation of the manuscript. Mosspiration Biotech provided support in the form of part of the salaries for authors [HB and HTS], but did not have any additional role in the study design, data collection and analysis, decision to publish, or preparation of the manuscript. The specific roles of these authors are articulated in the 'author contributions' section.

**Competing interests:** Author HB and HTS was partly employed at Mosspiration Biotech. This does not alter our adherence to PLOS ONE policies on sharing data and materials. The remaining authors declare that the research was conducted in the absence of any commercial or financial relationships that could be construed as a potential conflict of interest.

being toxic to the cell [9, 10]. Thus, an alternative way to produce and store compounds inside expression platforms was investigated.

We explored a novel approach and targeted terpenoids into plant lipid droplets (LDs) that was recently tested by transient expression in *Nicotiana benthamiana*. However, an algae lipid droplet protein was used, and the results were not conclusive possibly due to the change in carbon flow [11]. Plant cytosolic LDs are dynamic organelles that originate from the endoplasmic reticulum (ER) [12–14]. LDs consist of a hydrophobic core surrounded by a mono-lipid-layer membrane and structural proteins that maintain the membrane stability of LDs [15]. *P. patens* LDs are found in the vegetative gametophytes and are abundant in the spores. LDs store triacylglycerol (TAG) and steryl esters (SE) in a liquid form as an energy source [16]. The lipophilic environment attracts small hydrophobic compounds produced in its proximity. Since LDs are naturally buoyant, they will upon lysis of the cell separate from the rest by centrifugation, a property that lowers the downstream purification cost. In this work, we overexpressed patchoulol synthase (PTS), which is a sesquiterpene synthase that catalyze the production of patchoulol from farnesyl diphosphate in the cytosol. PTS and patchoulol originates from the scrub *Pogostemon cablin* [2]. We also overexpressed PTS physically attached to three LD related proteins, which are a part of the LD biogenesis (oleosin, seipin and lipid droplet associated protein also known as small rubber particle protein) to produce the sesquiterpenoid patchoulol close to LDs. These three groups of proteins were selected based on their diverse structure and LD binding properties.

Oleosins have been found to be the most abundant protein in the *P. patens* LD proteome [16]. They are attached to the cytosolic side of the ER and here support a hairpin loop on the membrane surface that leads to the emerging LDs [16, 17]. The support of the hairpin loop supports sequestration of neutral lipids from the ER to form the LD [18]. Following this sequestration oleosins together with other proteins supports the detachment of the LDs from the ER and release into the cytosol [18]. Here oleosins supports the structure of the LDs and hinder a formation of large LDs [14].

Seipin proteins are known to play a structural role in LD biogenesis in yeast and *Arabidopsis* [19, 20]. Seipins have multiple membrane-spanning helices that localize the protein to the ER membrane. Although the function of seipins is unclear due to conflicting results among different organisms [21], seipins have shown to effect the accumulation of TAG in LDs of *Arabidopsis* [19]. Currently, no work has been performed for seipins in non-vascular plants.

Unlike oleosin and seipin, Lipid Droplet Associated Protein (LDAP) (also known as small rubber particle protein (SRPs)) does not contain any transmembrane domain. LDAPs are mainly folded as a helical protein with self-polymerization properties that is capable of covering the LD surfaces without the disruption of the LD membrane structure [22, 23]. LDAPs are stress-related and required for proper growth in most plants and are found in cells both with and without LDs [14]. Previous works in *Arabidopsis* and tobacco have shown that *At*LDAP1 is localized to the LDs by covering the membrane surface of the LD [24–26].

Here, we also verified the localization of oleosin and LDAP1 to the LDs and seipins to the ER tubules in *P. patens*. Furthermore, by utilizing these LD-associated proteins as an anchor, we showed that the expression of LD protein-attached patchoulol synthase (PTS) led to a higher content of patchoulol in the LD as compared to the expression of free PTS.

## Materials and methods

### Plant material, growth conditions, and transformation

Wild type *Physcomitrium patens* (syn. *Physcomitrella patens*, Gransden ecotype) was obtained from the International Moss Stock Center at the University of Freiburg

(http://www.moss-stock-center.org/). Growth conditions and transformation processes are similar to those previously published [27].

## Strain construction

Transformation of *P. patens* was performed according to the protocol previously published [27–29] Approximately, 1.5 g (fresh weight) *P. patens* cells cultivated for five days were digested with 20 mL of 0.5% DriselaseR enzyme in 8.5% D-mannitol for 45 minutes with gentle shaking. Digested cells, filtered through a 100 μm pored mesh, were pelleted by centrifugation for 4 minutes at 150 x g. The pelleted protoplasts were washed and pelleted twice with a wash solution (8.5% D-mannitol, 10 mM $CaCl_2$). Washed protoplasts were concentrated to $1.6 \times 10^6$ protoplasts/mL and suspended in MMM solution (9.1% D-mannitol, 10% MES, and 15 mM $MgCl_2$). Protoplasts were transformed by mixing 300 μL of protoplast with an equal amount of PEG and 5μg of total DNA. The mixture was incubated for 5 minutes at 45˚C followed by another 5 minutes at room temperature. PEG in the mixture was diluted by adding 300 μL of 8.5% D-mannitol five times with 1 minute intervals in between, followed by adding 1 mL of 8.5% D-mannitol five more times. Transformed protoplasts were pelleted by centrifugation and the resulting pellet was resuspended in 500 μL of 8.5% D-mannitol and 2.5 mL of PRMT. The cells were spread on multiple PRMB solid media plates with a cellophane overlay. The cells were incubated overnight in a dark chamber and moved to continuous light for 5–7 days at 23˚C. The regenerated protoplasts were transferred to PhyB media containing the appropriate selection marker for 2- weeks. After two weeks of selection, the cellophane was transferred for two more weeks on PhyB media and back on to selection for two weeks. Stable transformants were genotyped by PCR and transferred into PhyB media. A detailed description of this protocol is published in [28, 30]. Above transformation method yielded the following stable moss mutants for this experiment (for sequences see below): ZmUbi:PpOle1, ZmUbi:PpOle1-Venus, ZmUbi:PpOle1-PTS, ZmUbi:PpOle1-LP4/2A-PTS, ZmUbi:PpSeipin325, ZmUbi:PpSeipin325-Venus, ZmUbi:PpSeipin325-PTS, ZmUbi:AtLDAP1, ZmUbi:AtLDAP1-Venus, ZmUbi:PpLDAP1-PTS, ZmUbi:PTS (also see S1 Table in S1 File). Coding sequences of all the DNA fragments were inserted into the *P. patens* neutral 108 loci, with G418 selection cassette, under the Maize Ubiquitin 1 promoter and OCS terminator (S1 Fig in S1 File). The primers used are listed in the S2 Table in S1 File.

## DNA fragments and genes

*PpSeipin325* (*Pp1S325_14V6.1*) and *PpOleosin1* (*Pp1S84_138V6.1*) (named PpOle1) was amplified from *P. patens* cDNA and chosen based on previous work on expression data and presence in all tissues of development [16]. The coding sequence of *Arabidopsis thaliana* lipid droplet associated protein 1 (AtLDAP1, At1g67360) and oleosin attached to PTS via LP4/2A linker was obtained using Thermofisher DNA synthesis services. The patchoulol synthase (PTS, AAS86323) was amplified using the pUNI33 PTS plasmid [2], and LP4/2A was obtained from DNA fragments previously shown to be functional in *P. patens* [30]. The G418 selection cassette and Maize Ubiquitin 1 promoter was amplified from the pMP1355 vector [30].

## Microscopy

LDs were stained using either 0.5 μg/ml of BODIPY 505/515 (Invitrogen Molecular Probes) or LipidTOX™ Red Neutral Lipid Stain 577/609 (ThermoFisher). Images of BODIPY stained LDs were obtained with a Leica Las AF confocal laser microscope using a 488 nm laser excitation. A window of 510–530 nm was used for LD visualization and a window of 650–700 nm window was used to visualize the auto-fluorescence of chloroplasts. For co-localization studies, the

yellow fluorescent protein Venus was excited with a 514-laser line and visualized with a window of 520–550. The LD staining dye HCS LipidTOX™ Red Neutral Lipid Stain was excited with the 594-laser line and visualized with a window of 605–630 nm. For the visualization of the ER, ER-Tracker™ Red (BODIPY™ TR Glibenclamide) dye was used. The emission and excitation parameters for ER-Tracker™ Red were similar to HCS LipidTOX™ Red Neutral Lipid Stain. Z-stacks were performed on all images with a line average of four. All the Z-stacks were combined using the Leica maximum projection function and Imaris software and processed using Imaris 9.1 (Oxford Instruments) and ImageJ [31] data analysis software. The size of the LDs was quantified by imaging with at least ten random images of multiple cells from biological triplicates. See Figs 1–3 for examples on how the pictures was zoomed in for the measurement, all pictures for LD size measurements is given in S3 File. The diameter of each non-overlapping LDs was measured using ImageJ and the size of the LDs was calculated using the volume of a sphere ($V = 4/3\ \pi r^3$).

## Metabolite extraction and GC-MS analysis

Biological triplicates of *P. patens* lines were grown for three weeks in 20 mL liquid PhyB (BCDAT) media under standard conditions, after which the tissues were washed, isolated and ground for patchoulol quantification inside the cells. For patchoulol quantification in the *P. patens* lines were extracted with an equal amount of 100% ethyl acetate by sonication for one hour. For patchoulol quantification outside of the cells, 10 mL of media was extracted with 2mL of 100% ethyl acetate after rapid shaking for 10 minutes. After phase separation 200 μL of

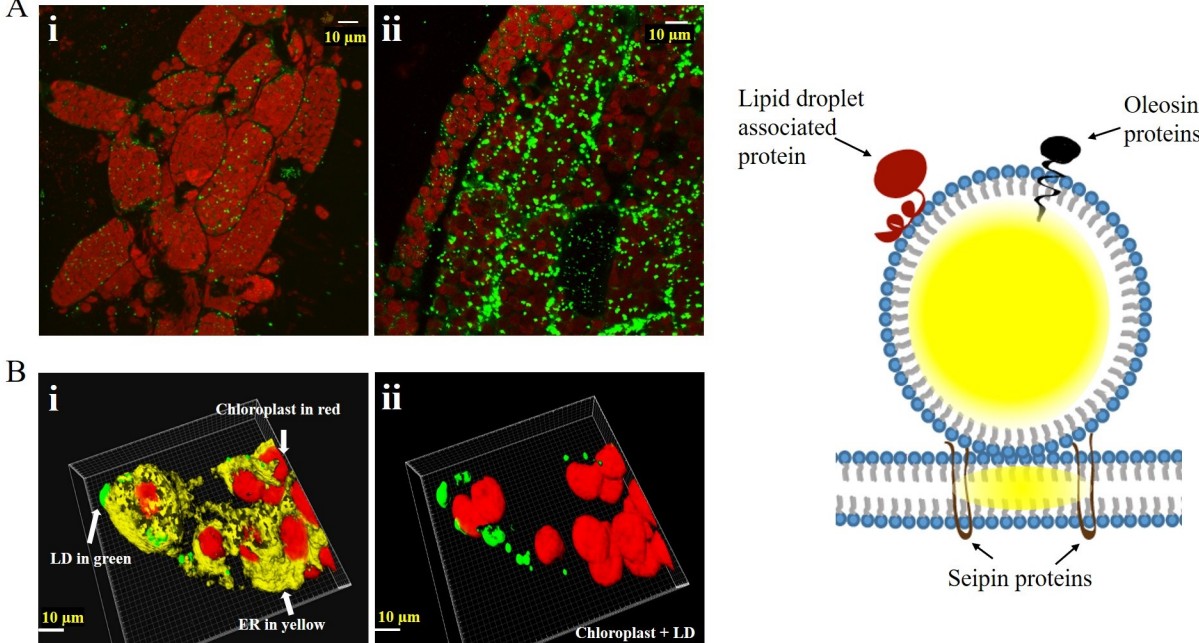

**Fig 1. Wild type LD staining.** (A) Microscopy images of WT protonema cells grown in for 10 days (i) and WT gametophytes grown for 6 weeks (ii). Stained with BODIPY (green) and overlaid with chloroplast autofluorescence (red), showing the presence of cytosolic LDs. (B) (i) Image depicts 3D projection of surface rendering Z-stack image of a portion of a cell stained with BODIPY and ER-Tracker™ Red dye was used. and overlaid with red chloroplast autofluorescence (ii) BODIPY stained LDs in green overlaid with red chloroplast autofluorescence after removing the ER tracker stained Z-stack image. The right-hand side shows a model for the localization of LD-associated proteins. Seipin oligomers localize in the ER at the ER-LD contact site. Lipid droplet associated proteins mainly fold as a helical protein and covers the LD surface without disturbing the LD membrane. Oleosin proteins penetrate the LD membrane into the LD matrix with a long conserved hydrophobic hairpin structure.

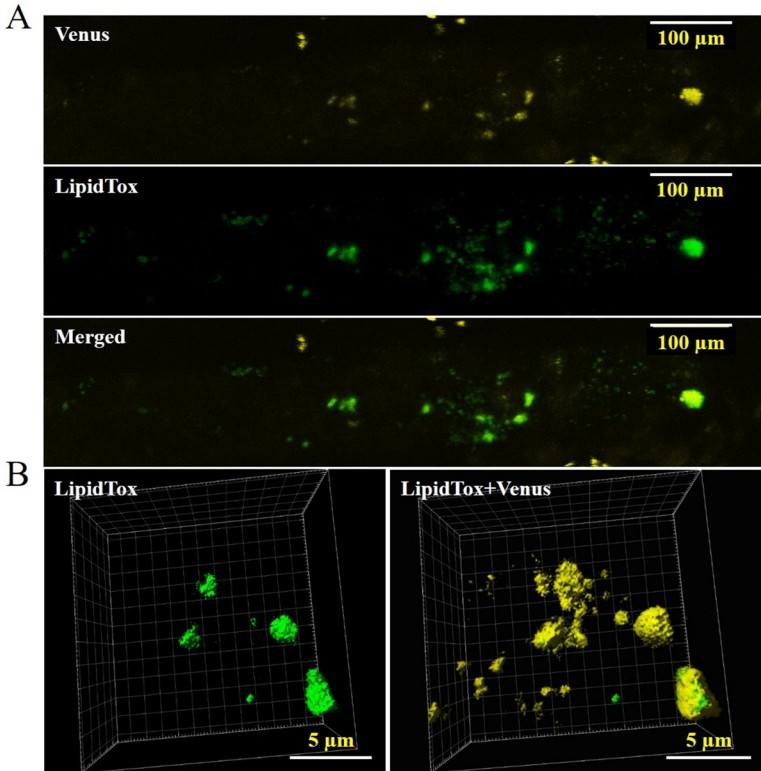

**Fig 2. Co-localization of PpOLE1 to LDs of *P. patens*.** (A) PpOLE1 was visualized by the fluorescence of the C-terminal tagged Venus and LDs were visualized by LipidTOX Red stain (artificial yellow color). (B) 3D projection of surface rendering Z-stacks of LipidTOX Red stained LDs (artificial yellow color) were merged with Venus for the depiction of LD co-localization)).

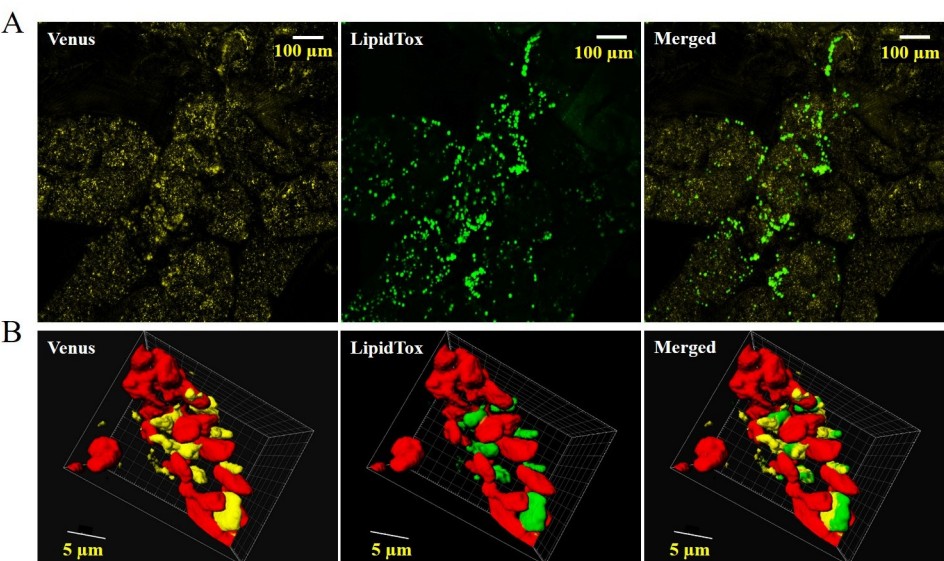

**Fig 3. Co-localization of AtLDAP1 to *P. patens* LDs.** A) Processed confocal Z-stacks of Venus tagged AtLDAP1 (yellow), LipidTOX red-stained LD (green) and subsequent overlap of the two micrographs (scale = 10 µm). B) 3D projection of surface rendering Z-stack image of the stained LDs (green) with chloroplast auto-fluorescent (red) and subsequent overlap of Venus localization of AtLDAP1.

the organic phase was loaded into GC vials and 1 μL was injected into the GC-MS. Patchoulol was quantified by using a standard curve (see S2 Fig in S1 File) of the authentic patchoulol standard, and the final concentration of patchoulol per dry weight was calculated. GC-MS analysis was performed on a GC-MS equipped with a ZB-5ms column (30 m x 0.25 mm x 0.25 μm) as previously described [2]. The samples were injected on a non-split mode using the following program: The initial temperature of the oven was held for 1 minute at 50˚C. Then, the temperature was increased to 320˚C at 15˚C/min and held at 320˚C for 5 minutes. The total run took 24 minutes. The injector temperature was set at 250˚C, a temperature that can lead to rapid degradation with sesquiterpenoids [32]. This was however not the case and the downstream analysis was not affected [32]. The ion source temperature of the mass spectrometer was at 230˚C. The obtained data obtained were compared with the standard for verification and quantification purposes.

## PTS expression analysis by RT-PCR

RNA transcription level for patchoulol synthase was measured by extracting the total RNA from 7-day-old cells, using Spectrum™ Plant Total RNA Kit (Sigma, STRN250). A total of 500 ng of each RNA sample was reverse transcribed to cDNA with the iScript™ cDNA Synthesis kit (BIO-RAD). RNA extraction was performed on biological triplicates for each *P. patens* mutant line. RT-PCR was performed using QuantiFast® SYBR® Green PCR (Qiagen) according to the manufacturer's protocol. The following amplification program was used: 95˚C 5 min, 40 cycles at 95˚C 10 s followed by 60˚C 30 s. Samples were amplified in duplicates from the same RNA isolation. RT-PCRs were performed using three biological replicates with qPCR primers designed for PTS, actin, and tubulin (S2 Table in S1 File). RT-PCR efficiency, E, was estimated for each gene by generating standard curves by plotting quantification cycle (Cq) values (y) against the log of a serial cDNA dilution (x). For this, a cDNA sample was used as template in a range of 25, 5, 2.5, 1.2 and 0.6 ng. The RT-PCR efficiencies were calculated from the slope of the linear regression equations of the standard curves, along with the regression coefficient ($R^2$). The equation used was: $E = 10(-1/a)$, E values in a range of 1.90–2.10 (PCR efficiency between 90% and 110%) with a regression coefficient below 0.02 are acceptable. All PCR efficiencies displayed between 100% and 115%.

## Growth measurement

Plant height was measured on biological triplicates of wild type (WT), and the *ZmUbi*:*AtL-DAP1-PTS* and *ZmUbi*:*PTS* lines. These were grown for 20 days in BCD liquid media. Cell lines were imaged using 5X bright field microscopy by attaining at least 15 random images of each cell line. The plant height of all the gametophytes with a minimum of two nodes in the image was calculated using ImageJ software by drawing a segmented line along the length of the stem manually. Measuring tool is freely available in the Image J software.

## Lipid droplet extraction

Protoplasts of biological triplicates were isolated according to a previously published protocol [28] from cells that were cultivated in PhyB liquid media for three weeks. Washed protoplasts were lysed through osmotic pressure by adding 10 mL of deionized water. Lysed cells were pelleted at 5,800 x g for 10 minutes and the supernatant was mixed with sucrose to a final concentration of 0.3 M. The mixture was loaded into an ultracentrifuge tube with a 2 mL 150 mM KCl layer on top and centrifuged for 40 minutes at 135,000 x g in a swinging bucket rotor. The top 1.5 ml of the KCl layer containing the LDs were isolated using a glass tube and transferred into another ultracentrifuge tube for further purification. The isolated 1.5 mL of 150 mM KCl

with LDs was carefully overlaid with a 10.5 mL of 10 mM KCl followed by 2 mL of diH$_2$O. This mixture was further centrifuged by ultracentrifugation using the same settings as above. Purified LDs on the top of the water layer were confirmed by BODIPY staining. The purified LDs were extracted with ethyl acetate and the extracts were analyzed using GC-MS. The GC-MS analysis was performed on a Thermo Scientific TSQ 8000 Evo with a TriPlus RSH Autosampler equipped with agitation oven and needle bake out option. The injection port on the GC was set at 200˚C, with a splitless time of 3 min. The flow through the column was 1.0 mL/min using Helium as carrier gas. The column used was a 30m DB-5MS column, 0.250 mm i.d., 0.25 μm film thickness. The oven temperature program was set to 60˚C for 3 min, raised at 10˚C/min to 300˚C and a hold time of 3 min. The MS Ion source temperature was 200˚C, transfer line temperature was 250˚C, and the scan from m/z 50 to m/z 300 using 70 eV electrical ionization with a 5 min. solvent delay. Data and spectra were analyzed using Thermo Scientific Tracefinder software version 4.1 and the NIST library database version NIST 14 and compared to previously published data [2].

## RNA extract for sequencing

The total RNA was extracted using the Spectrum™ Plant Total RNA Kit (Sigma, STRN250) from cells grown in PhyB media for 7 days after blending as described for RT-PCR. This yielded 300 μg total RNA as determined by nano-drop. RNA integrity was initially confirmed by agarose gel electrophoresis and the visualization of intact ribosomal RNA bands. Subsequent RNA quality control was carried out on a 2100 Bioanalyzer (Agilent Technologies, Hørsholm, Denmark) and each sample received an RNA integrity numbers (RIN) of greater than 8.5.

The total RNA of biological triplicates was pooled to make one technical sample, and total RNA samples were submitted to Macrogen (Seoul, Korea) for stranded mRNA library preparation using an Illumina Truseq Stranded mRNA library prep kit. The sequencing library is prepared by random fragmentation of the DNA or cDNA sample, followed by 5' and 3' adapter ligation. Alternatively, "tagmentation" combines the fragmentation and ligation reactions into a single step that greatly increases the efficiency of the library preparation process. Adapter-ligated fragments were then PCR amplified and gel purified. For cluster generation, the library was loaded into a flow cell where fragments were captured on a lawn of surface-bound oligos complementary to the library adapters. Each fragment was then amplified into distinct, clonal clusters through bridge amplification. When cluster generation was complete, the templates were ready for sequencing. The sequencing was performed using Novaseq 150bp paired-end sequencing providing 30–40 million reads per sample.

Illumina SBS technology utilizes a proprietary reversible terminator-based method that detects single bases as they are incorporated into DNA template strands. As all 4 reversible, terminator-bound dNTPs are present during each sequencing cycle, natural competition minimizes incorporation bias and greatly reduces raw error rates compared to other technologies.

Sequencing data was converted into raw data for the analysis. The raw reads were uploaded to the Sequence Read Archive (SRA) at NCBI with the Bioproject accession number PRJNA603435.

## Trinity assembly and transcriptome annotation

A *de novo* transcriptome was assembled as a reference for read mapping and Differential Expression (DE) analysis using Trinity v2.4.0 [33]. As recommended in the Trinity protocol, one single Trinity assembly was generated by combining all reads across samples as input to ease following downstream analysis. Quality trimming and adapter removal was performed

using trimmomatic [34] with default parameters. Transcript abundance was estimated using the alignment-based quantification method RSEM that uses Bowtie2 [35] as an alignment method. Transcript and gene expression matrices were generated, and the numbers of expressed genes were calculated. Finally, differential expression analysis was performed at the gene level using edgeR [36] with a dispersion rate of 0.1. Extractions and clustering of differentially expressed genes were performed with combinations of P-value cutoff for FDR of 1e-3 and fold-change values of 2, 4, 16, 64 and 256. Functional annotation of the transcriptome was performed using the annotation suite Trinotate. The functional annotation includes homology searches to BLAST, SwissProt, PFAM and various annotation databases such as eggNOG/GO/ Kegg.

## Results

### Lipid droplet formation in wild type *P. patens*

To understand the LD biogenesis during default condition, we observed the formation of LDs in two different developmental stages in *P. patens* wild type. When compared to the protonema cells where LDs were sparsely scattered, we could observe a substantial increase in the numbers of LD in mature leafy gametophytes (Fig 1A). This is in line with previous studies that showed the relationship between LD biogenesis and senescence [37, 38]. The processed confocal images of BODIPY and ER-tracker stained tissue suggested that the LDs in *P. patens* originated from the ER (Fig 1B), as previously observed [16]. In the combined images of chloroplasts and LDs (average size of 3.12 $E^{-11}$ ± 6.21$E^{-16}$ mm$^3$ (Table 1, S4 Fig in S1 File), it was observed that LD seems to be positioned on the outer edges of the ER adjacent to the chloroplasts (Fig 1A and 1B). This indicate that LD formation in *P. patens* is similar to that of higher plants. The abundance of LD scaffolds in mature cells of *P. patens* showed the feasibility of targeting PTS to LDs for stable storage of patchoulol *in vivo*. For this purpose, we selected three LD-associated proteins due to their diverse way of association to LDs (Fig 1C).

### Localization of oleosin, LDAP and Seipin in *P. patens* leafy gametophyte tissues

To test if the LD related proteins, PpOLE1, PpSeipin325, AtLDAP1 are localized to the LDs of *P. patens*, the proteins were tagged with the reporter protein Venus using a flexible linker to minimize the steric effects of the two proteins. Confocal images of the leafy gametophyte tissues showed that PpOLE1 is specifically localized to the LDs (Fig 2), which confirms previous findings of the localization of oleosin proteins to the LD in *P. patens* gametophyte [16].

**Table 1. LD size of different *P. patens* lines (based on biological triplicates).**

| Cell type | LD Diameter (mm) | Volume (mm$^3$) | Fold increase of volume |
|---|---|---|---|
| WT | 4.73E-04 ± 1.28E-05 | 3.12E-11 ± 6.21E-16 | - |
| *PTS*-only | 5.65E-04 ± 1.30E-05 | 5.30E-11 ± 6.49E-16 | 1.70 [*] |
| *PpOle1-LP4/2A-PTS* | 5.49E-04 ± 1.13E-05 | 4.88E-11 ± 4.28E-16 | 1.56 [*] |
| *PpSeipin325* OE | 6.60E-04 ± 1.49E-05 | 8.45E-11 ± 9.65E-16 | 2.71 [**] |
| *PpSeipin325-Venus* | 5.99E-04 ± 1.30E-05 | 6.33E-11 ± 6.40E-16 | 2.03 [**] |
| *PpSeipin325-PTS* | 6.74E-04 ± 1.32E-05 | 9.01E-11 ± 6.77E-16 | 2.88 [**] |

The size of the LDs was assessed by a combination of random confocal images and ImageJ analysis of the LDs of biological triplicates. More than 10 different random micrographs and at least 200 LDs were analyzed per cell type. The significance of the data was determined by one-way ANOVA test comparing the *P. patens* lines with the WT; $P < 0.05$ = [*], $P < 0.005$ = [**].

LDAP is one of the most abundant proteins in *Arabidopsis* leaf-LD proteome and shown to control the LD biogenesis in leaves [39]. The *Arabidopsis* genome contains three *LDAP* genes, but *P. patens* only contains one *LDAP* homolog (*PpLDAP*). Here AtLDAP1 was investigated as this was already characterized and had shown to be involved in LD stability [39]. According to the Affymetrix GeneChip data, *PpLDAP* is expressed in gametophytes and showed an increased expression in reproductive stages like the other LD-proteins (S5 Fig in S1 File). As seen in the confocal micrographs, AtLDAP1 is localized to the LDs of *P. patens* (Fig 3), this confirms the possible association of LDAP to LDs in *P. patens*. The result suggests that LD protein interaction is highly conserved in the evolutionary lineage, from *P. patens* to higher plants. Compared to the WT, the *ZmUbi:AtLDAP1* line was observed to produce LDs in excess. Due to the frequent overlapping of the LDs in the processed confocal Z-stack images, the average LD number per cell was not obtained. Further studies are required to conclude the effect and the function of LDAPs on LD biogenesis in *P. patens*.

The localization of seipins was confirmed by examination of PpSeipin325 (one of two seipins in *P. patens*, Fig 4, S6 Fig in S1 File). Composite confocal images of the *ZmUbi:PpSeipin325-Venus* line showed that the majority of PpSeipin325 follows the ER pattern, which confirms previous findings in other organisms [19, 20]. Further image processing, using Imaris software showed that LDs emerge from a tiny round clearing of the seipin expression pattern (Fig 3).

## Patchoulol accumulation in *P. patens* with PTS and oleosin overexpression

Previous work has demonstrated that 0.2–0.8 mg/g DW patchoulol could be produced in *P. patens* by overexpressing *PTS*, with higher concentration by plastid targeting and

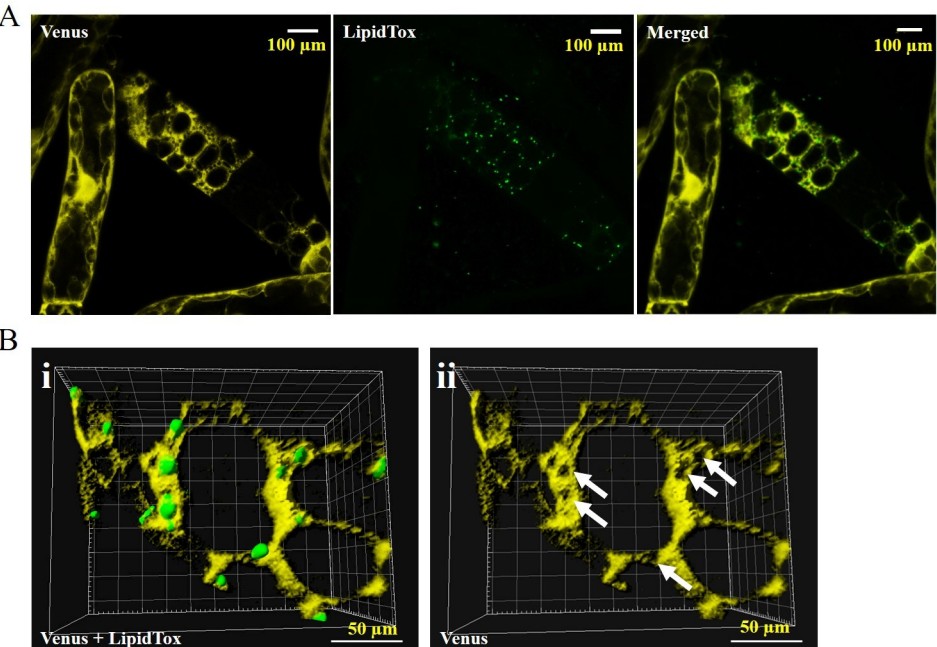

**Fig 4. Localization of PpSeipin325 in *P. patens*.** A) PpSeipin325 was tagged at the C-terminal with Venus protein (yellow) and imaged for the localization using confocal microscopy. LDs were stained with LipidTOX red (artificial green color) and merged with the Venus expression. Merged image depicts that the majority of LDs (green) are following the expression pattern of PpSeipin325 (yellow). (B) (i)3D projection of surface rendering Z-stack PpSeipin325 (yellow) with LDs (green). (ii) PpSeipin 325 expression pattern (yellow) after removing the LD Z-stack image. Arrows indicated the holes where the LDs (green) are emerging (scale = 10 μm).

overexpression of FPP synthase [2]. In the current work, the *ZmUbi:PTS* line produced 32.89 µg/g of patchoulol, retaining 10.07 ± 0.42 µg/g DW inside the cell and 22.82 ± 1.18 µg/g DW excreted to the media (Table 2). Although this is lower than the previously published result, in this work PTS synthase was targeted to the *P. patens* neutral 108 locus with a single copy integration, unlike the random and multiple integrations of *PTS* in the previously published work [2].

To address the question of the effect of LD-anchoring on patchoulol production, PTS was linked to PpOLE1 by two different linkers: Flexible Gly-Ser (GS) linker for physical attachment and translationally cleaved LP4/2A linker [27, 40]. Unlike the GS linker, the LP4/2A linker facilitates the release of PTS from PpOLE1 upon cleavage by protease, this at least at 80% cleavage efficiency [40, 41], allowing us to directly compare the retention of patchoulol inside the cells with or without the LD anchor.

As seen in Table 2, the physically linked PTS (*ZmUbi:PpOle1-PTS*) retained the most patchoulol inside the cell, amounting to 43% of the total patchoulol production, where the translationally cleaved *ZmUbi:PpOle-LP42A-PTS* retained only 27% and the PTS only *ZmUbi:PTS* retaining 31% (Table 2). Taken together, the proximity of PTS to PpOLE1 aided the retention of patchoulol inside the cells. We also confirmed the occurrence of patchoulol and the two minor biosynthesis products, seychelene and patchoulene, in the LDs by the GC-MS analysis of the purified LDs of *ZmUbi:PpOle1-PTS* line (S2 Fig in S1 File).

Gentle osmolytic lysis of protoplast and a thorough wash step was attempted to prevent any ER membrane residues or carry-over patchoulol from the cytoplasm, to the LD fraction. Other bryophyte species, such as *Marchantia polymorpha* L, can accumulate sesquiterpenoids in LDs along with neutral lipids as previously observed [42, 43]. Following the cell lysis, only the free LDs were isolated, and not the LDs still physically attached to the ER. As shown in Table 1 the isolated free-LDs in the *ZmUbi:PTS* line was larger than in WT, where biosynthesis of patchoulol in the cell could have had an effect on the increased LD size. Over-expression of PTS and oleosin (*ZmUbi:PpOle-LP4/2A-PTS*) as translationally cleaved proteins produced smaller LD compared to *ZmUbi:PTS* line. Overexpression of oleosin proteins has previously been shown to decrease the size of LDs in seeds [44]. Thus, the LD volume reduction in *ZmUbi:PpOle-LP4/2A-PTS* compared to *ZmUbi:PTS* could be partially explained by the functional properties of the oleosin protein that lead to more but smaller LDs.

The occurrence of LDs in each of the *ZmUbi:PTS*, *ZmUbi:PpOle-PTS* and *ZmUbi:PpOle-LP4/2A-PTS* lines were comparably higher than in WT cells (Fig 5). The increased number of LDs in *ZmUbi:PpOle-PTS* and *ZmUbi:PpOle-LP4/2A-PTS* lines were expected due to the known function of oleosin. However, the increased number of LDs in *ZmUbi:PTS* line was unexpected. This observation of an increased number of LDs was also documented in artemisinin *P. patens* mutants produced in a previous work [45].

**Table 2. Patchoulol production of three weeks old modified *P. patens* cell lines (biological triplicates).**

| Cell Type | Inside the cell (µg/g) | Outside the cell (µg/g) | Total Production (µg/g) | % Inside |
|---|---|---|---|---|
| *PpOle1-PTS* | 1.86 ± 0.30 | 2.43 ± 0.17 | 4.3 | 43 [*] |
| *PpOle1-LP4/2A-PTS* | 4.67 ± 0.30 | 12.63 ± 2.26 | 17.3 | 27 |
| *PpSeipin325-PTS* | 1.81 ± 0.34 | 3.13 ± 0.72 | 4.95 | 37 |
| *AtLDAP1-PTS* | 5.14 ± 0.79 | 7.41 ± 0.89 | 12.54 | 41 [*] |
| *PTS*-Only | 10.07 ± 0.42 | 22.82 ± 1.18 | 32.9 | 31 |

Patchoulol was extracted from cells (inside) and the media (outside) using ethyl acetate. Production was calculated via GC/MS analysis of the samples. The amount of patchoulol inside the cell was divided by the total amount to calculate the percentage of patchoulol contained in the cells. The significance of the data was determined by student T-test comparing the retained patchoulol in *P. patens* lines with the PTS alone line; P < 0.05 = *.

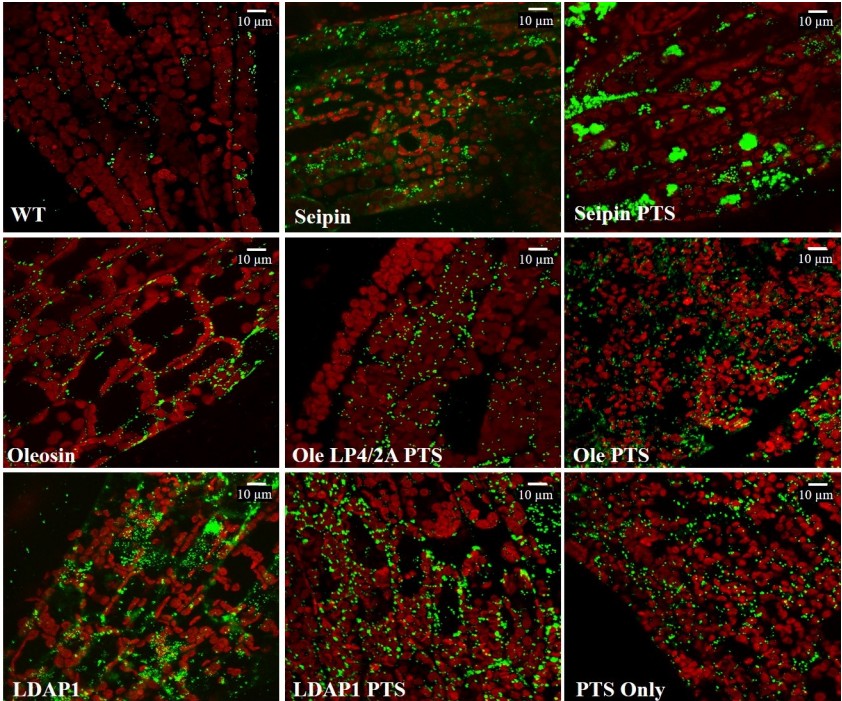

**Fig 5. Ectopic expression of LD-associated proteins and PTS promoted LD occurrence.** Representative confocal Z-stack images of LDs in gametophyte leaves of different *P. patens* lines. Green color shows the BODIPY stained LDs and the red shows the chloroplast autofluorescence. Images were collected at the same laser intensity and magnification (scale = 10 µm).

## Patchoulol accumulation and growth effects in PTS linked AtLDAP1

To gain insight into the effects of LDAP linked PTS, *ZmUbi:AtLDAP1-PTS* was stably transfected to *P. patens*. After 3 weeks, the *ZmUbi:AtLDAP1-PTS* line produced a total of 12.5 µg/g DW patchoulol (Table 2). The *ZmUbi:AtLDAP1-PTS* line produced the highest amount of patchoulol compared to any LD targeted PTS line. The high patchoulol production of the *ZmUbi:AtLDAP1-PTS* line could be attributed to the structural association of AtLDAP1 to the LDs. The unlikely presence of any transmembrane domains in the LDAP protein structure possibly increased the folding efficacy of attached PTS, to yield a high production of patchoulol. Overall, the retention of patchoulol inside the cells of the *ZmUbi:AtLDAP1-PTS* line was comparable to the translationally cleaved *ZmUbi:Oleosin-LP4-2A-PTS* line. The *ZmUbi:AtLDAP1-PTS* line retained 41% patchoulol inside the cell as compared to 28% of the *ZmUbi:Oleosin-LP4/2A-PTS* line (Table 2). The *ZmUbi:AtLDAP1-PTS* line also harbored a large number of LDs similar to that of *ZmUbi:Oleosin-LP4/2A-PTS* (Fig 5). The size of the LDs in the *ZmUbi:AtLDAP1-PTS* was not measured as the LDs clumped together and was impossible to seperate. This subset of data re-validates the significance of having the PTS in close proximity to LD for storage purposes and depicts the importance of the structural properties of the LD-anchor for patchoulol biosynthesis.

## Patchoulol production in PpSeipin325-PTS mutants and size of LD

PTS was also attached to the PpSeipin325, due to the unique association of seipin proteins to the LD expansion on the ER (Fig 1C). In principle, the *ZmUbi:PpSeipin325-PTS* line could function from the outside of the emerging LD, aiding the expansion of the LD with neutral

lipids and patchoulol. The engineered *ZmUbi:PpSeipin325-PTS*, produced a total of 4.95 µg/g patchoulol in 3 weeks (Table 2). The very low production of patchoulol in *PpSeipin325-PTS* line could partially be explained by a very low expression of PTS in the *ZmUbi:PpSeipin325-PTS* line. This could be due to the large size of PpSeipin325, which subsequently leads to lower expression and likely alters the folding and properties of the PTS and thereby the kinetics (S3 Fig in S1 File).

To further demonstrate the transfer and storage of patchoulol to LDs, we examined the expansion of the LD size in the *ZmUbi:PpSeipin325-PTS* cell line relative to that of *ZmUbi:PpSeipin325-Venus*. The outside association of seipin protein made the *ZmUbi:PpSeipin325-PTS* cell line an ideal candidate to study the LD expansion in relation to patchoulol production. The *ZmUbi:PpSeipin325-Venus* cell line was selected as the control to have similar biological conditions of having an attached protein to seipin. The *ZmUbi:PpSeipin325-Venus* line produced LDs of 6.33E-11 ± 6.4E-16 mm$^3$ in size as compared to the *ZmUbi:PpSeipin325-PTS* that produced a LD of 9.01E-11± 6.77E-16 mm$^3$. On average, LDs of *ZmUbi:PpSeipin325-PTS* line expanded by 188% to the WT and 42% to the *ZmUbi:PpSeipin-Venus* control (Table 1, S4 Fig in S1 File).

Our confocal micrographs suggest that *P. patens* seipins are localized mainly to the ER and around the base of the LDs. Work done on *Arabidopsis* seipins has shown that AtSeipin1 is capable of producing bulky LDs [19]. Interestingly, larger LDs were observed in the *ZmUbi:PpSeipin325* line in 3-week-old leaf tissues when compared to WT. On average, the LDs in the *ZmUbi:PpSeipin325* line was 2.71 fold larger than wild type LDs (Table 1, S4 Fig in S1 File). This suggests that similar to AtSeipin1, PpSeipin325 is involved in the expansion of the LDs.

### Effect on plant growth

*P. patens* expressing only PTS had 26% lower growth in plant height when compared to WT after 20 days, measured in millimeter of the gametophyte growth. However, producing patchoulol with PTS attached to LD-associated proteins had no negative effect on the growth but provided an even faster biomass development than WT. This was observed for the *ZmUbi:AtLDAP1-PTS* when the plant heights were compared to WT, and *ZmUbi:PTS* lines. The average WT gametophyte grown for 20 days, grew to 0.32 ± 0.01 mm in height, followed by 0.24 ± 0.01mm for *ZmUbi:PTS* and 0.65 ± 0.03 mm for *ZmUbi:AtLDAP1-PTS* cell lines. The *ZmUbi:AtLDAP1-PTS* line had a 2-fold plant-height increase over the WT.

### Transcriptomic sequencing data

Due to the unexpected increased number of LDs in *ZmUbi:PTS* line, transcriptome analysis for each mutant was performed. The transcriptomic data obtained showed that among the genes upregulated 256-fold or more, two *P. patens* genes were present in all our samples (Fig 6, Table 3, summary in S2 File). This was GDSL esterase/lipase At4g16230-like (XP_024368984) and non-specific lipid-transfer protein 3-like (XP_024363543). A third lipase, the GDSL esterase/lipase At5g03820-like (XP_024382012), is represented in all but the *Pp*Seipin325 overexpression line. More lipases are also upregulated, and genes related to photosynthesis is downregulated along with other genes in all overexpression lines. In order to confirm the involvement of these genes in LD biogenesis further studies are needed.

## Discussion

Our results from Table 1 demonstrate the relationship between LD expansion and expression of PTS in the cells. While the results are not conclusive for routing of patchoulol into LDs, they do demonstrate that increasing the size and numbers of LDs in the cell will retain more

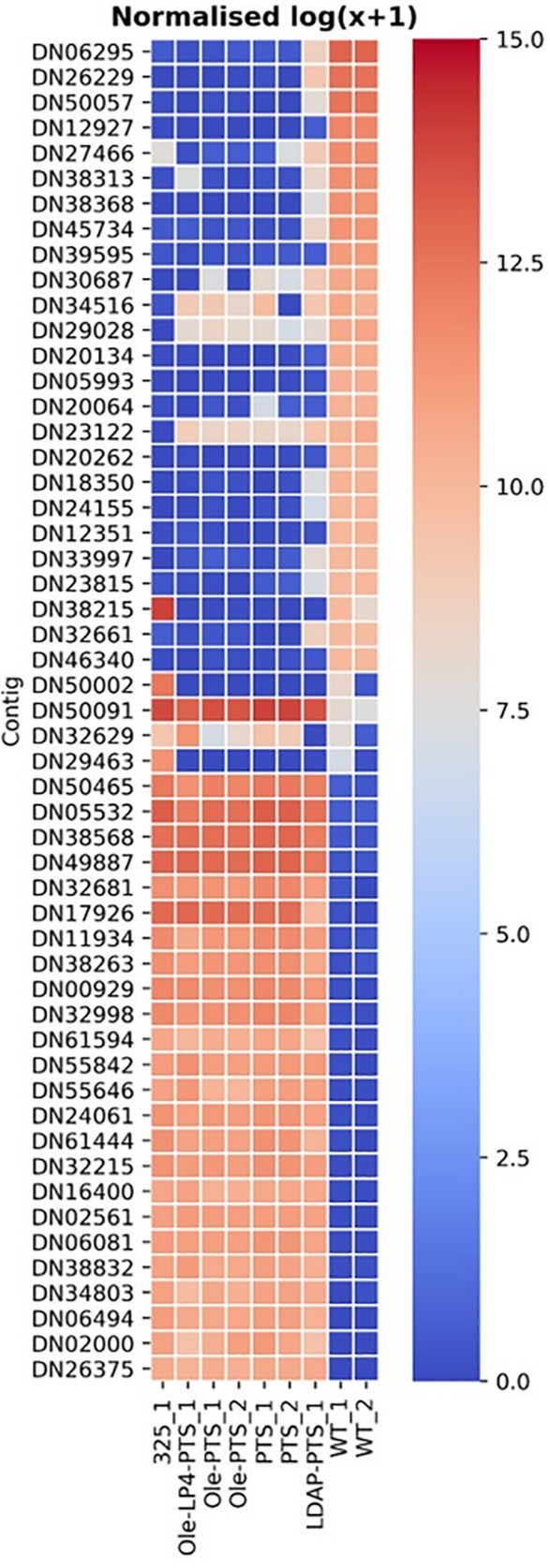

**Fig 6. Log (x+) normalized heatmap of the different *P. patens* lines and the wildtype *P. patens* with a 256-fold up (red) or down (blue) regulation of gene expression in one or more *P. patens* lines compared to the wild type.** For some this also show that there is little to no change in one or more *P. patens* lines, but a 256-fold change in just one *P. patens* line. The annotation of the contig numbers is seen in Table 3.

patchoulol inside the cells and the LDs. The level of patchoulol biosynthesis is not as high in the PTS-LD linked mutants as the two mutants where it is not linked. This also fits with what has previously been observed in *N. benthamiana* [11] and likely arise from a lower enzyme activity of PTS when the enzyme is attached to the LD's. The difference between the PTS alone and PpOle1-LP4/2A-PTS is probably due to change in transcription/translation of the longer construct; however, this is not examined.

We suggest that newly synthesized patchoulol in the PpSeipin325-PTS, AtLDAP1-PTS and PpOle1-PTS lines could enter the LD via two routes: passive diffusion the LD through the membrane in the cytoplasm or across the ER membrane into the ER tubule and then enter the LD through the ER-LD contact site. The latter is particularly interesting for the PpSeipin325-PTS, whereas for LDAP and Ole it is more likely that it is passive diffusion directly into LD. It is unknown if one route is preferred over the other, though both routes requires a passive diffusion over a lipid membrane.

Microscopic analysis verified the localization of PpOLE1 and AtLDAP1 to the cytosolic LDs. Over-expression of both *PpOle1* and *AtLDAP1* increased the number of cytosolic LDs compared to the WT. Localization of PpSeipin325 was restricted to the reticular ER pattern and over-expression of *PpSeipin325* increased the size of the LDs by 170%. Thus, we hypothesized that PpSeipin325, similar to AtSeipin1, is involved in the expansion of LDs. Large groups of clustering LDs were observed in the leaf tissue of three weeks old mutant line (Fig 5). LD structural proteins such as oleosins and LDAPs have the ability to coat the LDs and prevent the LDs from aggregation. The rapid expansion of the LDs in the *ZmUbi:PpSeipin325-PTS* line could hinder the natural localization process of LD structural proteins, leading to clustered LDs. Overall, an increase in LD numbers seems to provide a new pool for storage of sesquiterpenoids. Future work could focus on co-expressing PpSeipin325 with PpOLE1 or AtLDAP1 to attain a homogeneous pool of LDs to store patchoulol with expression of PTS.

As previously shown, the heterologous expression of terpenoids leads to growth deficiencies in cells [9, 10, 46]. It was observed that the production of terpenoids in *P. patens* led to a 48% lower biomass formation during 18 days of cultivation [46]. Here, *P. patens* expressing only PTS had 26% lower growth in plant height when compared to WT after 20 days. This stress response is probably also related to the enhanced LD production, which are formed to lower the stress burden to the cells. This stress response hypothesis is also supported by the enhanced growth in mutants where PTS is attached to the LDs thus ensuring that patchoulol is captured inside the LDs. A 2-fold increase in plant height of the *ZmUbi:AtLDAP1-PTS* is in line with previous observations where the over-expression of *AtLDAP1* in *Arabidopsis* increased the height by 5-fold [24]. How the LDAP proteins are involved in tissue growth in *Arabidopsis* and *P. patens*, is yet to be investigated, but the increase in growth and biomass overall lead to higher production of the final product. From the transcriptomic data in Table 3, it is clear that several lipases and non-specific lipid transporter proteins (LTPs) are upregulated in all the mutant lines. This indicates that the increased lipid droplet formation also initiates a higher production and transport of lipids. Our data supports that there is tight link between neutral lipid levels, LD formation and terpenoid production capacity. This link is supported by the recent study that showed the biogenesis of LD in chloroplast with modified oleosin remarkably increased the terpene production [47]. Previously published work in some bryophytes also suggest that LTPS are involved in the trafficking of fatty acids and larger lipids [48–50]. Thus,

**Table 3.** List of contigs shown in Fig 3.

| Contig number in sequencing assembly and general regulation (U/D/N) | *P. patens* NCBI # | Description from *P. patens* annotation |
|---|---|---|
| DN00929 (U) | XP_024395254 | chalcone synthase 6-4-like |
| DN02000 (U) | PNR47760 | hypothetical protein |
| DN02561 (U) | XP_024385936 | pectinesterase 15 -like |
| DN05532 (U) | XP_024380033 | alpha carbonic anhydrase 7-like |
| DN05993 (D) | N/A | uncharacterized protein |
| DN06081 (U) | XP_024395980 | peroxidase 21-like isoform X1 |
| DN06295 (D) | XP_024375887 | chlorophyll a-b binding protein of LHCII type 1-like |
| DN06494 (U) | XP_024382012 | GDSL esterase/lipase At5g03820-like |
| DN11934 (U) | XP_024368003 | uncharacterized protein |
| DN12351 (D) | XP_024367311 | copper transporter 5.1-like |
| DN12927 (D) | XP_024362229 | germin-like protein 9–3 |
| DN16400 (U) | XP_024395390 | receptor-like protein 51 |
| DN17926 (U) | XP_024368032 | alpha carbonic anhydrase 5-like |
| DN18350 (D) | XP_024397347 | enolase 1-like |
| DN20064 (D) | XP_024392911 | protein LURP-one-related 15-like |
| DN20134 (D) | XP_024403288 | polyphenol oxidase-like |
| DN20262 (D) | PNR36153 | hypothetical protein |
| DN23122 (N) | XP_024378832 | dirigent protein 19-like |
| DN23815 (D) | XP_024360335 | 1-Cys peroxiredoxin-like |
| DN24061 (U) | XP_024368984 | GDSL esterase/lipase At4g16230-like |
| DN24155 (D) | PNR55678 | hypothetical protein |
| DN26229 (D) | XP_024400099 | copper chaperone for superoxide dismutase |
| DN26375 (U) | XP_024382195 | uncharacterized protein |
| DN27466 (D) | XP_024399688 | phospho-2-dehydro-3-deoxyheptonate aldolase 2 |
| DN29028 (N) | N/A | uncharacterized protein |
| DN29463 (D) | NP_904206.1 | PSII 44 kD protein |
| DN30687 (D) | N/A | uncharacterized protein |
| DN32215 (U) | PNR26297 | UDP-glycosyltransferase 83A1-like |
| DN32629 (N) | N/A | uncharacterized protein |
| DN32661 (D) | PNR52172 | hypothetical protein |
| DN32681 (U) | PNR53501 | hypothetical protein |
| DN32998 (U) | XP_024363339 | abscisic acid 8'-hydroxylase 3-like |
| DN33997 (D) | N/A | uncharacterized protein |
| DN34516 (N) | XP_024359073 | chlorophyll a-b binding protein |
| DN34803 (U) | XP_024369838 | uncharacterized protein |
| DN38215 (D) | N/A | uncharacterized protein |
| DN38263 (U) | XP_024381396 | uncharacterized protein |
| DN38313 (D) | XP_024360503 | GDSL esterase/lipase At4g01130-like |
| DN38368 (D) | PNR36232 | hypothetical protein PHYPA_022083 |
| DN38568 (U) | XP_024393797 | uncharacterized protein |
| DN38832 (U) | PNR53501 | hypothetical protein |
| DN39595 (D) | XP_024358907 | formin-1-like |
| DN45734 (D) | PNR34789 | hypothetical protein |
| DN46340 (D) | PNR36032 | hypothetical protein |
| DN49887 (U) | XP_024393312 | aldehyde oxidase GLOX-like |
| DN50002 (D) | NP_904175.1 | PSII 47kDa protein |
| DN50057 (D) | XP_024367800 | ferric reduction oxidase 6-like |

(*Continued*)

**Table 3.** (Continued)

| Contig number in sequencing assembly and general regulation (U/D/N) | *P. patens* NCBI # | Description from *P. patens* annotation |
|---|---|---|
| DN50091 (U) | PNR39672 | hypothetical protein |
| DN50465 (U) | XP_024363543 | non-specific lipid-transfer protein 3-like |
| DN55646 (U) | XP_024375379 | non-specific lipid-transfer protein 2G-like |
| DN55842 (U) | XP_024377855 | probable xyloglucan endotransglucosylase/hydrolase |
| DN61444 (U) | XP_024388184 | pathogen-related protein-like |
| DN61594 (U) | XP_024393924 | germin-like protein 1–1 |

The up (U) or down (D) regulation or no change (N) is given for the majority of *P. patens* lines as compared to the expression in the wild type. The annotation provided is based on a BLAST search towards the *P. patens* genome v. 3.3, and the NCBI numbering is included.

the increased lipid biosynthesis and lipid content in the cell could provide better storage capacity for terpenoids. It requires further studies to prove this possibility, but this holds the potential for novel biotechnological developments.

## Conclusion

We observed 26% less growth in plant height when patchoulol is biosynthesized in *P. patens*, just by over-expressing *PTS*. This finding was in line with similarly published works, however, in the LD targeted PTS line, *ZmUbi:AtLDAP1-PTS* we observed 2-fold increased growth in plant height compared to WT. This shows that we can biosynthesize terpenoids in *P. patens*, increase growth, and potentially device a new extraction method, which is very useful for future industrial production.

All the LD protein-PTS attached lines retained more patchoulol in the cell, over none-attached *ZmUbi:PTS* and *ZmUbi:PpOle1-LP4/2A-PTS* lines. The size and number of LDs in patchoulol biosynthesis lines were increased; the size was increased with up to 188% and about 25% more patchoulol was retained inside the cell, though the production was significantly reduced. Thus, there is a potential to use the LDs for storage of small molecules during production in bioreactors. However, it requires further work to be established this for production of any small molecule.

## Supporting information

**S1 File. The file contains the following data.**
(PDF)

**S2 File. Differential expressed genes in lipid bodies.** Genes 256-fold differentially expressed between the mutants are shown in tables of the individual mutants.
(PDF)

**S3 File. Images used for measurement of LD size, and the measurements in excel sheets.**
(ZIP)

## Author Contributions

**Conceptualization:** Anantha Peramuna, Henrik Toft Simonsen.

**Data curation:** Anantha Peramuna, Hansol Bae, Carmen Quiñonero López, Arvid Fromberg, Bent Petersen, Henrik Toft Simonsen.

**Formal analysis:** Anantha Peramuna, Hansol Bae, Carmen Quiñonero López, Arvid Fromberg, Bent Petersen, Henrik Toft Simonsen.

**Funding acquisition:** Henrik Toft Simonsen.

**Investigation:** Anantha Peramuna, Bent Petersen, Henrik Toft Simonsen.

**Methodology:** Anantha Peramuna, Hansol Bae, Arvid Fromberg, Henrik Toft Simonsen.

**Project administration:** Henrik Toft Simonsen.

**Resources:** Hansol Bae, Henrik Toft Simonsen.

**Supervision:** Henrik Toft Simonsen.

**Validation:** Hansol Bae, Carmen Quiñonero López, Henrik Toft Simonsen.

**Visualization:** Hansol Bae, Henrik Toft Simonsen.

**Writing – original draft:** Anantha Peramuna, Henrik Toft Simonsen.

**Writing – review & editing:** Anantha Peramuna, Hansol Bae, Carmen Quiñonero López, Bent Petersen, Henrik Toft Simonsen.

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
