## [Decision Letter · Decision Letter 0]

13 Oct 2020

PONE-D-20-29050

Connecting moss lipid droplets to patchoulol biosynthesis

PLOS ONE

Dear Dr. Simonsen,

Thank you for submitting your manuscript to PLOS ONE. After careful consideration, we feel that it has merit but does not fully meet PLOS ONE’s publication criteria as it currently stands. Therefore, we invite you to submit a revised version of the manuscript that addresses the points raised during the review process.

Three scientists, who are each experts in specific aspects of the manuscript and at different academic stages, have kindly provided specific input. All three were very supportive, an opinion which I can only second. No additional experiments were suggested and the data was considered scientifically sound. However, there are suggestions and comments regarding the text, figures, minor details, and in particular one part, which appears to have been accidentally copied from another paper. Overall, I hope this will allow you to make a swift revision.

We look forward to receiving your revised manuscript.

Kind regards,

Björn Hamberger

Academic Editor

PLOS ONE

Journal Requirements:

2.Thank you for stating the following in the Acknowledgments Section of your manuscript:

[AP, HB, and HTS were supported by The Danish Council for Independent Research (#4005-00158B).

516 CQ was supported by funding from the People Program (Marie Curie Actions) of the European

517 Union's Seventh Framework Program FP7/2007-2013/ under REA grant agreement n° [607011].]

 [No - The funders had no role in study design, data collection and analysis, decision to publish, or preparation of the manuscript.]

3.Thank you for stating the following in the Competing Interests section:

[I have read the journal's policy and the authors of this manuscript have the following competing interests:

Author HB and HTS was partly employed at Mosspiration Biotech. The remaining authors declare that the research was conducted in the absence of any commercial or financial relationships that could be construed as a potential conflict of interest.].   

We note that one or more of the authors are employed by a commercial company: Mosspiration Biotech, Hørsholm, Denmark

Reviewers' comments:

Reviewer's Responses to Questions

**Comments to the Author**

1. Is the manuscript technically sound, and do the data support the conclusions?

Reviewer #1: No

Reviewer #2: Yes

Reviewer #3: Yes

2. Has the statistical analysis been performed appropriately and rigorously? 

Reviewer #1: I Don't Know

Reviewer #2: Yes

Reviewer #3: Yes

3. Have the authors made all data underlying the findings in their manuscript fully available?

Reviewer #1: No

Reviewer #2: Yes

Reviewer #3: Yes

4. Is the manuscript presented in an intelligible fashion and written in standard English?

Reviewer #1: Yes

Reviewer #2: Yes

Reviewer #3: Yes

5. Review Comments to the Author

Reviewer #1: Review Connecting moss lipid droplets to patchoulol biosynthesis

In this manuscript, the authors have targeted a patchoulol synthase to lipid droplets (LD) or the ER by fusing it to different LD/ER proteins. The targeting appears to be successful. The goal to increase the production of patchoulol in comparison to previous studies where patchuolol synthase was not targeted to LDs was not reached, as the levels that could be obtained were much lower than previously described. The targeting in this study had however a benefit to reach higher levels. Furthermore, the LD size was slightly increased by the overexpression. While the authors did not get very far in a biotechnological sense, I like the idea of using LDs as sites for the synthesis and storage of hydrophobic compounds.

Comments

1. In the abstract the authors write:

“Ectopic expression of PTS increased the number and size of LDs, implying an unknown mechanism between heterologous terpene production and LD biogenesis.”

However, there could also be an effect on LD degradation. Also, the number of LDs was not quantified. It is furthermore strange that in Figure 1 the number of LDs in the wild type is very high and then in Figure 5 much lower. Where all the images taken at the same time of the day or the plants fixed? LDs undergo a circadian rhythm.

2. It would be interesting to see examples, how the LD size was determined. I don’t think that LD size can be reliably measured by confocal microscopy as they are too small. I mean again it would be great to see the original pictures they took, but I would assume that the LDs are only a few pixels in diameter. Furthermore, often LDs seem to be clustered together and single LDs cannot be resolved.

Calculating volumes from this data to enlarge the differences is not necessary. The data should be presented in a box plot or better a swarm plot so that individual sizes can be seen.

3. In the lines 51-53 reviews on plant LDs should be cited instead or at least additionally. Paper number 14 for example does not mention plants at all.

4. The authors should at least briefly describe the reaction catalyzed by PTS and where it is normally located and what the substrate is.

5. Line 62 “Oleosins are the most abundant protein in the P. patens LD proteome”. On what publication is this based on? Was it really quantified? Is this true for spores only or in all moss tissues?

6. Lines 64-66. “Oleosins sequestrate neutral lipids from the ER bilayer and this way extract the LDs from the ER. This extraction of LD to the cytosol is controlled solely by the innate properties of oleosin, and modified oleosin can redirect the LD to the ER lumen and then to the vacuoles”.

While these sentences are a 1:1 plagiate from Anthony Huangs review, I feel that this statement is also misleading. I don’t think the role of oleosins in LD formation is so clear. They are definitely not needed in plants and also the absence of proteins including VAP27-1, Seipin, LDAP and LDIP have a strong influence on the proper formation of LDs indicating that it is not oleosin alone that does the job in vivo.

7. The authors should discuss that the accumulation of LDs could also just be a stress symptom as the plants are clearly stressed by PTS overexpression as they grow smaller. I don’t think that the small amount of patchoulol produced can really substantially contribute to LD formation. If the authors want to make this claim or indicate this possibility they must measure the amount of TAG and sterol esters to show that these do not increase.

Reviewer #2: The present study by Peramuna et al investigates synthetic biology approaches to co-produce high value sesquiterpene (patchoulol) together with three different Lipid droplet proteins Oleosin (PpOLE1), Lipid Droplet (LD) Associated Protein or Seipin (PpSeipin325) for redirecting the patchoulol production in Physcomitrella patens. The study involves overexpressing patchoulol synthase (PTS) physically attached to three proteins to produce the sesquiterpenoid patchoulol close to LDs. The authors also verified the localization of PpOLE1 and AtLDAP1 to the LDs and PpSeipin325 to the ER tubules in P. patens. By utilizing these LD-associated proteins as an anchor, the authors showed that the expression of LD protein-attached patchoulol synthase (PTS) led to a higher content of patchoulol in the LD as compared to the expression of free PTS.

The study in principle is sound. The experiments are well done, and the data generated by the authors appears to be of good quality. The study lays a groundwork for better targeted metabolic engineering of P. patens for terpene production in future. The manuscript is a great fit for the journal and hence is recommended for acceptance once the following comments and queries are addressed:

Comments

Line 40-41. Please add the reference for ….. “fluctuating market price of 30-200 US dollars/Kg.”

Line Line 48-51. The statement needs further refinement. The use of term “novel” and “the results were not conclusive” is somewhat misleading as the use of both LDAPs and Oleosins for terpene compartmentalization has been well demonstrated previously [ref 11 and 47]

Results:

P. patens has three oleosin genes, OLE1-OLE3, and OLE1 and OLE2 has two splice variants each [Ref. 15]. The authors do not explain why OLE1 was used for the present study and also which splice variant of OLE1 was used?

Line 291- 294: the supplementary figure mentions PpSRP instead of PpLDAP. Please correct this.

Line 257: Please check and correct the order in which supplementary figures are mentioned in the text.

Line 288-296: Although it does not matter from the synthetic biology point of view to use AtLDAP1 instead of PpLDAP, it would have been more useful to test the biological function of PpLDAP and compare it with AtLDAP1. I do not suggest the authors to do it for the current manuscript, but it is surely something they can do for future work, given the robustness of their moss transgenic system and the expertise in the field.

Line 301-305: The authors do not explain why PpSeipin325 was chosen for the study, since there are two of them present in P. patens? Were these already characterized in the past? If these were, please mention that in the text. If not, the rationale for not using PpLDAP does not hold that well. The authors are advised to strengthen this section of the manuscript.

Line 327: The authors do not explain why they chose a neutral locus Pp108 instead of random and multiple integration locus, when the production was demonstrated to be higher in their previous research?

Line 335: What is the difference between using a free PTS and one tagged to Oleosin via LP4/2A linker, because principally PTS will be free in the cytosol in both the cases? Also, why was not LP4/2A linker tested with AtLDAP1 and Seipin325 constructs?

Line 407: replace “tested” with “observed”

Line 423-431: The authors talk about the effect of the different combinations of lipid droplets/PTS on the P. patens growth. But there are no figures/pictures to support the findings. I think this is very important observation and some representative images/figure in the main text would be quite valuable.

Line 433: Although the transcriptome data seems to be a detailed study on its own and could have been very useful in understanding the reasons for unexpected increase of LDs in some of the lines, it does not seem to be the case here. I would suggest the authors to put more details in this section. The authors can use some of the bioinformatics tools to provide a chart or a diagram of major metabolic pathways that are affected in these lines.

Conclusion: The conclusion sections needs some refinement. It just seems like an extension of results/ discussion.

Figures:

1. Please label the subcellular organelles in the microscopy images.

2. If possible, add the corresponding phase contrast images of the microscopy images.

3. Add a comparative plant growth figure of wild and transgenic lines.

Legends:

Line 444-448: Please add a description of the transgenic lines in the legend section.

Reviewer #3: This review is being written from the perspective of a senior PhD candidate with background in Physcomitrella patens, terpene biosynthesis, and molecular biology and expertise in bioinformatics, genetics, and genomics. Peremuna et al. clearly express their teams attempts to increase yield of patchoulol and localize PTS production to the ER and more specifically lipid droplets. When compared to previously published papers their work appeared to have mixed successes, with lower patchoulol productions then in other lines, which they account for by the location of gene insertion being in a neutral locus. Additionally, lines targeted to lipid droplets showed less patchoulol production but better retention within the cell and healthier plant growth when compared to non-localized mutants.

I think the logic and science in this paper is sound and I very much so appreciated the transparency with the mixed success of patchoulol production and PTS integration into the lipid droplets. I also think this paper does a great job of bringing attention to the questions that arose throughout their scientific process that remained unanswered, posing future directions not only to themselves but the readers as well. Their work and the publishing of this work will likely save others time and energy in the future from having to repeat similar or the same experiments tested here.

I think this paper requires no major edits but could be improved by including more background on why a heterologous system is necessary in the first place. Also, be aware of the use of acronyms and abbreviations as they were not always as clear as they could be (ie. Line 29 – LD/ER refers to Lipid Droplet/endoplasmic reticulum but ER is not defined previously; Line 72 - PpOLE1/PpSeipin325/AtLDAP1 not being formally highlighted in the introduction; Table 2 - Possible typo stating PpLDAP1-PTS, instead of AtLDAP1-PTS (check between lines 382 and 684); Line 490 – Adding another acronym (LTPs) near the end of the paper and is only used one other time seems unnecessary). With a paper containing many acronyms and frequently referring to each of them in different orders in each section it is important to make sure the information is conveyed clearly.

Lastly, some constructive advice to the authors for their future work I have some suggestions regarding my expertise. (Line 235) In the future you may want to refer to the published genome and trancriptome of P. patens instead of recreating a de novo transcriptome with Trinity, even though these lines have had gene transformations, the universal gene names of P. patens would likely be helpful. Additionally, the version of trinity used is a few years old and many updates have occurred since this version (Version 2.11.0 is currently available). (Line 433) As you investigate more into the differential gene expression make sure to consider genes that are less than a 256-fold change in expression as well. 256-fold is a good place to start but there is probably a lot of useful information at less extremes as well. (Line 298) You mention in a few places that there was an observed increase in LD number per cell but you weren’t able to get a calculation on the actual number of droplets that changed. You may be able to use your average lipid droplet volume and average total lipid production to get an idea of how many droplets there were using the following equation and solving for DropletNumber LDVolume*NumberOfDroplets=TotalLipidContent?

6. PLOS authors have the option to publish the peer review history of their article (what does this mean?). If published, this will include your full peer review and any attached files.

Reviewer #1: No

Reviewer #2: **Yes: **Wajid Waheed Bhat

Reviewer #3: **Yes: **Davis T. Mathieu

---

## [Author Response · Author response to Decision Letter 0]

15 Oct 2020

Comments to the reviewers:

Reviewer 1

1. In the abstract the authors write:

“Ectopic expression of PTS increased the number and size of LDs, implying an unknown mechanism between heterologous terpene production and LD biogenesis.”

However, there could also be an effect on LD degradation. Also, the number of LDs was not quantified. It is furthermore strange that in Figure 1 the number of LDs in the wild type is very high and then in Figure 5 much lower. Where all the images taken at the same time of the day or the plants fixed? LDs undergo a circadian rhythm.

Answer: In figure 1 the LD’s was taken at day 10 and at 6 weeks. Figure 5 is after 3 weeks of growth. We did not observe that LD’s was influenced by circadian rhythms, as we the plants was grown un 24h light. Thus Figure 5 is best compared with Figure 1.Ai. Figure text for figure 5 is updated.

2. It would be interesting to see examples, how the LD size was determined. I don’t think that LD size can be reliably measured by confocal microscopy as they are too small. I mean again it would be great to see the original pictures they took, but I would assume that the LDs are only a few pixels in diameter. Furthermore, often LDs seem to be clustered together and single LDs cannot be resolved. 

Answer: We would like to refer the reviewer to see Figs 1, 2 and 3 for examples on how the pictures was zoomed in for the measurement. It is quite clear that some are bundles of LD’s where as others are clearly single LDs. This is also described in M&M, the text is updated and strengthened here.

Calculating volumes from this data to enlarge the differences is not necessary. The data should be presented in a box plot or better a swarm plot so that individual sizes can be seen.

Answer: We are not sure this would provide further evidence and support. And as such will keep the current representation.

3. In the lines 51-53 reviews on plant LDs should be cited instead or at least additionally. Paper number 14 for example does not mention plants at all.

Answer: the references are updated.

4. The authors should at least briefly describe the reaction catalyzed by PTS and where it is normally located and what the substrate is.

Answer: sure, this is added to the text.

5. Line 62 “Oleosins are the most abundant protein in the P. patens LD proteome”. On what publication is this based on? Was it really quantified? Is this true for spores only or in all moss tissues?

Answer: This is based on Huang C-Y, Chung C-I, Lin Y-C, Hsing Y-IC, Huang AHC. Oil Bodies and Oleosins in Physcomitrella Possess Characteristics Representative of Early Trends in Evolution. Plant Physiol. 2009;150: 1192–1203.

And yes it was quantified and it is not only in spores. We would like to refer the reviewer to this publication.

6. Lines 64-66. “Oleosins sequestrate neutral lipids from the ER bilayer and this way extract the LDs from the ER. This extraction of LD to the cytosol is controlled solely by the innate properties of oleosin, and modified oleosin can redirect the LD to the ER lumen and then to the vacuoles”.

While these sentences are a 1:1 plagiate from Anthony Huangs review, I feel that this statement is also misleading. I don’t think the role of oleosins in LD formation is so clear. They are definitely not needed in plants and also the absence of proteins including VAP27-1, Seipin, LDAP and LDIP have a strong influence on the proper formation of LDs indicating that it is not oleosin alone that does the job in vivo.

Answer: The text was updated to reflect more recent knowledge and the comment from the reviewers.

7. The authors should discuss that the accumulation of LDs could also just be a stress symptom as the plants are clearly stressed by PTS overexpression as they grow smaller. I don’t think that the small amount of patchoulol produced can really substantially contribute to LD formation. If the authors want to make this claim or indicate this possibility they must measure the amount of TAG and sterol esters to show that these do not increase.

Answer: This is added to the discussion. 

Reviewer 2:

Line 40-41. Please add the reference for ….. “fluctuating market price of 30-200 US dollars/Kg.”

Answer: Please see reference 2, where these numbers are given.

Line Line 48-51. The statement needs further refinement. The use of term “novel” and “the results were not conclusive” is somewhat misleading as the use of both LDAPs and Oleosins for terpene compartmentalization has been well demonstrated previously [ref 11 and 47]

Answer: the sentence was rephrased.

Results:

P. patens has three oleosin genes, OLE1-OLE3, and OLE1 and OLE2 has two splice variants each [Ref. 15]. The authors do not explain why OLE1 was used for the present study and also which splice variant of OLE1 was used?

Answer: The full native sequence of PpOleosin1 (Pp1S84_138V6.1) was used as given in M&M and chosen based on expression in all tissues. M&M is updated to reflect this. 

Line 291- 294: the supplementary figure mentions PpSRP instead of PpLDAP. Please correct this.

Answer: The text is updated accordingly.

Line 257: Please check and correct the order in which supplementary figures are mentioned in the text.

Answer: The text is updated accordingly.

Line 288-296: Although it does not matter from the synthetic biology point of view to use AtLDAP1 instead of PpLDAP, it would have been more useful to test the biological function of PpLDAP and compare it with AtLDAP1. I do not suggest the authors to do it for the current manuscript, but it is surely something they can do for future work, given the robustness of their moss transgenic system and the expertise in the field.

Answer: We fully agree, and hope to start this work soon.

Line 301-305: The authors do not explain why PpSeipin325 was chosen for the study, since there are two of them present in P. patens? Were these already characterized in the past? If these were, please mention that in the text. If not, the rationale for not using PpLDAP does not hold that well. The authors are advised to strengthen this section of the manuscript.

Answer: The full native sequence of PpSeipin325 (Pp1S325_14V6.1) was used as given in M&M and chosen based on expression in all tissues. M&M is updated to reflect this. 

Line 327: The authors do not explain why they chose a neutral locus Pp108 instead of random and multiple integration locus, when the production was demonstrated to be higher in their previous research?

Answer: Previous research have not examined how many copies was inserted into the genome. It is much likely that 5-8 copies of PTS were inserted previously. With a controlled 108 integration one know precisely what was done. There is not need to explain this.

Line 335: What is the difference between using a free PTS and one tagged to Oleosin via LP4/2A linker, because principally PTS will be free in the cytosol in both the cases? Also, why was not LP4/2A linker tested with AtLDAP1 and Seipin325 constructs?

Answer: The discussion is updated to reflect this.

Line 407: replace “tested” with “observed”

Answer: The text is updated accordingly.

Line 423-431: The authors talk about the effect of the different combinations of lipid droplets/PTS on the P. patens growth. But there are no figures/pictures to support the findings. I think this is very important observation and some representative images/figure in the main text would be quite valuable.

Answer: this data is represented in the tables. A figure cannot be produced.

Line 433: Although the transcriptome data seems to be a detailed study on its own and could have been very useful in understanding the reasons for unexpected increase of LDs in some of the lines, it does not seem to be the case here. I would suggest the authors to put more details in this section. The authors can use some of the bioinformatics tools to provide a chart or a diagram of major metabolic pathways that are affected in these lines.

Answer: The current discussion of the transcriptome data reflects what we find must valuable at the current state. Further analysis of these data would require further in vivo studies as well to cooperate the findings. Thus, we have chosen to represent that here, and share these data, without a too detailed discussion and conclusions that we find it hard to support.

Conclusion: The conclusion sections needs some refinement. It just seems like an extension of results/ discussion.

Answer: The text is updated accordingly.

Figures:

1. Please label the subcellular organelles in the microscopy images.

Answer: The text is updated accordingly and updated legend.

2. If possible, add the corresponding phase contrast images of the microscopy images.

Answer: we do not have this image.

3. Add a comparative plant growth figure of wild and transgenic lines.

Answer see comment above on the same issue.

Legends:

Line 444-448: Please add a description of the transgenic lines in the legend section.

Answer: The text is updated accordingly.

 

Reviewer 3

I think this paper requires no major edits but could be improved by including more background on why a heterologous system is necessary in the first place. Also, be aware of the use of acronyms and abbreviations as they were not always as clear as they could be (ie. Line 29 – LD/ER refers to Lipid Droplet/endoplasmic reticulum but ER is not defined previously; Line 72 - PpOLE1/PpSeipin325/AtLDAP1 not being formally highlighted in the introduction; 

Table 2 - Possible typo stating PpLDAP1-PTS, instead of AtLDAP1-PTS (check between lines 382 and 684); 

Line 490 – Adding another acronym (LTPs) near the end of the paper and is only used one other time seems unnecessary). With a paper containing many acronyms and frequently referring to each of them in different orders in each section it is important to make sure the information is conveyed clearly.

Answer: The text is updated accordingly.

Lastly, some constructive advice to the authors for their future work I have some suggestions regarding my expertise. (Line 235) In the future you may want to refer to the published genome and trancriptome of P. patens instead of recreating a de novo transcriptome with Trinity, even though these lines have had gene transformations, the universal gene names of P. patens would likely be helpful. Additionally, the version of trinity used is a few years old and many updates have occurred since this version (Version 2.11.0 is currently available). (Line 433) As you investigate more into the differential gene expression make sure to consider genes that are less than a 256-fold change in expression as well. 256-fold is a good place to start but there is probably a lot of useful information at less extremes as well. (Line 298) You mention in a few places that there was an observed increase in LD number per cell but you weren’t able to get a calculation on the actual number of droplets that changed. You may be able to use your average lipid droplet volume and average total lipid production to get an idea of how many droplets there were using the following equation and solving for DropletNumber LDVolume*NumberOfDroplets=TotalLipidContent?

Answer: we investigated the suggested methods and idea and will use in future studies. We tried to use the average volume but the data we had and the measurements was not really sufficient at this stage. Thus we have chosen to stick with what we have. But we are very glad for the advises. As mentioned above, a further analysis of the transcriptome data is for sure needed, however this is currently not possible for us at this stage, and would require further in vivo studies.

---

## [Decision Letter · Decision Letter 1]

12 Nov 2020

PONE-D-20-29050R1

Connecting moss lipid droplets to patchoulol biosynthesis

PLOS ONE

Dear Dr. Simonsen,

Thank you for re-submitting your manuscript to PLOS ONE. After careful consideration, we feel that it has significantly improved but does not fully meet PLOS ONE’s publication criteria as it currently stands. Therefore, we invite you to submit a revised version of the manuscript that addresses the points raised during the review process.

Next to minor text edits, which are all specified below, one remaining comment addresses the data availability for the quantification of the lipid droplets, which should be addressed. 

We look forward to receiving your revised manuscript.

Kind regards,

Björn Hamberger

Academic Editor

PLOS ONE

Reviewers' comments:

Reviewer's Responses to Questions

**Comments to the Author**

1. If the authors have adequately addressed your comments raised in a previous round of review and you feel that this manuscript is now acceptable for publication, you may indicate that here to bypass the “Comments to the Author” section, enter your conflict of interest statement in the “Confidential to Editor” section, and submit your "Accept" recommendation.

Reviewer #1: (No Response)

Reviewer #2: All comments have been addressed

Reviewer #3: All comments have been addressed

2. Is the manuscript technically sound, and do the data support the conclusions?

Reviewer #1: Partly

Reviewer #2: Partly

Reviewer #3: Yes

3. Has the statistical analysis been performed appropriately and rigorously? 

Reviewer #1: I Don't Know

Reviewer #2: Yes

Reviewer #3: Yes

4. Have the authors made all data underlying the findings in their manuscript fully available?

Reviewer #1: No

Reviewer #2: No

Reviewer #3: Yes

5. Is the manuscript presented in an intelligible fashion and written in standard English?

Reviewer #1: Yes

Reviewer #2: Yes

Reviewer #3: Yes

6. Review Comments to the Author

Reviewer #1: 1. I still believe that if a statement is made about the number of LDs these should be quantified with sufficient numbers of biological replicates and images.

Please provide all the images taken for measurement of number and LD size in the supplements according to the PLoS ONE policy. Also please give all the individual datapoints of all experiments in supplemental table as outlined in the PLOS data policy.

https://journals.plos.org/plosone/s/data-availability

5. Line 62 “Oleosins are the most abundant protein in the P. patens LD proteome”. On what publication is this based on? Was it really quantified? Is this true for spores only or in all moss tissues?

Answer: This is based on Huang C-Y, Chung C-I, Lin Y-C, Hsing Y-IC, Huang AHC. Oil Bodies and Oleosins in Physcomitrella Possess Characteristics Representative of Early Trends in Evolution. Plant Physiol. 2009;150: 1192–1203.And yes it was quantified and it is not only in spores. We would like to refer the reviewer to this publication

Reply: Thank you for pointing out this publication. It should be cited right after this statement not only after the next statement. Furthermore, in this paper only one tissue was investigated. Therefore, the statement should be refined and made more specific. Also the statement itself is based on a blue gel, where one band was further investigated by MALDI-MS. A quantification is something different to me. But ok.

6. Thank you for amending the statement but the new statement

“Following this sequestration Oleosins together with other proteins supports the detachment of the LDs from the ER and release into the cytosol.”

Could use a citation for example a recent review or the new Vap27 paper by the chapman group.

Reviewer #2: The revised version of the manuscript by Peramuna et al does answer most of the comments/queries raised by the reviewers, and with some minor revisions, the manuscript is deemed fit for acceptance in PLOS One. The comments are as under:

1. Line 70: change “Oleosins to “oleosins”.

2. Line 354: Please clarify the statement “Gentle lysis of protoplast with osmosis and….”

3. Line 408: replace “properly” with “probably”

4. Line 460: replace “with” with “between”

5. Line 463-464: Rephrase the sentence “The amount of production is not as high in the……..”.

6. Line 5465, 466 and 489: “properly” with “probably”

7. Line 490-491: please rephrase the sentence “This is supported by that attaching………….”

8. Line 501-502: rephrase “Previously published works also suggest that some bryophyte LTPs are involved….” to “Previously published work in some bryophytes also suggest that LTPS are involved….”

9. Line 502-503: Please rephrase the sentence “ Thus, with increased lipid biosynthesis and more lipids in the cell could provide…….”.

Reviewer #3: All comments I made from first review were adequately answered. Upon reviewing second submission, the paper has made excellent revisions in the introduction and results section, adding to clarity and necessity of research. No Further major edits necessary.

Minor edits to consider after my second reading:

Line 60 - Sentence regarding the fragrance of patchoulol is unnecessary for the understanding of the paper as a whole. Authors choice to leave it in or not

Line 280 - typo (show  shows)

Line 396 - "The size of the LDs in the ZmUbi:AtLDAP1-PTS was not measured due to their direct binding to the LDs.": This sentence was unclear to me. Is "their" referring to the protein? Did you not measure lipid droplet size because of direct binding of AtLDAP1-PTS to LDs? Wouldn't a larger lipid droplet support your argument that it was correctly targeted to the LDs.

Line 464 - "This also fits with what was observed in N. benthamiana [11] and properly arise from a change in the kinetics for the PTS to yield lower turnover numbers when this attached to the LD's": Consider rewording this sentence, Current wording is confusing and/or grammatically incorrect(?)

7. PLOS authors have the option to publish the peer review history of their article (what does this mean?). If published, this will include your full peer review and any attached files.

Reviewer #1: No

Reviewer #2: **Yes: **Wajid Waheed Bhat

Reviewer #3: **Yes: **Davis T. Mathieu

---

## [Author Response · Author response to Decision Letter 1]

17 Nov 2020

Reviewer #1: 1. I still believe that if a statement is made about the number of LDs these should be quantified with sufficient numbers of biological replicates and images.

Please provide all the images taken for measurement of number and LD size in the supplements according to the PLoS ONE policy. Also please give all the individual datapoints of all experiments in supplemental table as outlined in the PLOS data policy.

https://journals.plos.org/plosone/s/data-availability

comments for the author, including concerns about dual publication, research ethics, or publication ethics. (Please upload your review as an attachment if it exceeds 20,000 characters)

Reviewer #1: 1. I still believe that if a statement is made about the number of LDs these should be quantified with sufficient numbers of biological replicates and images.

Please provide all the images taken for measurement of number and LD size in the supplements according to the PLoS ONE policy. Also please give all the individual datapoints of all experiments in supplemental table as outlined in the PLOS data policy.

https://journals.plos.org/plosone/s/data-availability

Answer: The data is provided as extra supplementary files.

5. Line 62 “Oleosins are the most abundant protein in the P. patens LD proteome”. On what publication is this based on? Was it really quantified? Is this true for spores only or in all moss tissues?

Answer: This is based on Huang C-Y, Chung C-I, Lin Y-C, Hsing Y-IC, Huang AHC. Oil Bodies and Oleosins in Physcomitrella Possess Characteristics Representative of Early Trends in Evolution. Plant Physiol. 2009;150: 1192–1203.And yes it was quantified and it is not only in spores. We would like to refer the reviewer to this publication

Reply: Thank you for pointing out this publication. It should be cited right after this statement not only after the next statement. Furthermore, in this paper only one tissue was investigated. Therefore, the statement should be refined and made more specific. Also the statement itself is based on a blue gel, where one band was further investigated by MALDI-MS. A quantification is something different to me. But ok.

Answer: The reference was added and the text was slightly altered.

6. Thank you for amending the statement but the new statement

“Following this sequestration Oleosins together with other proteins supports the detachment of the LDs from the ER and release into the cytosol.”

Could use a citation for example a recent review or the new Vap27 paper by the chapman group.

 The reference was added and the text was slightly altered.

Reviewer #2: The revised version of the manuscript by Peramuna et al does answer most of the comments/queries raised by the reviewers, and with some minor revisions, the manuscript is deemed fit for acceptance in PLOS One. The comments are as under:

1. Line 70: change “Oleosins to “oleosins”.

2. Line 354: Please clarify the statement “Gentle lysis of protoplast with osmosis and….”

3. Line 408: replace “properly” with “probably”

4. Line 460: replace “with” with “between”

5. Line 463-464: Rephrase the sentence “The amount of production is not as high in the……..”.

6. Line 5465, 466 and 489: “properly” with “probably”

7. Line 490-491: please rephrase the sentence “This is supported by that attaching………….”

8. Line 501-502: rephrase “Previously published works also suggest that some bryophyte LTPs are involved….” to “Previously published work in some bryophytes also suggest that LTPS are involved….”

9. Line 502-503: Please rephrase the sentence “ Thus, with increased lipid biosynthesis and more lipids in the cell could provide…….”.

Answer: All the corrections has been made in the manuscript.

Reviewer #3: All comments I made from first review were adequately answered. Upon reviewing second submission, the paper has made excellent revisions in the introduction and results section, adding to clarity and necessity of research. No Further major edits necessary.

Minor edits to consider after my second reading:

Line 60 - Sentence regarding the fragrance of patchoulol is unnecessary for the understanding of the paper as a whole. Authors choice to leave it in or not

Line 280 - typo (show  shows)

Line 396 - "The size of the LDs in the ZmUbi:AtLDAP1-PTS was not measured due to their direct binding to the LDs.": This sentence was unclear to me. Is "their" referring to the protein? Did you not measure lipid droplet size because of direct binding of AtLDAP1-PTS to LDs? Wouldn't a larger lipid droplet support your argument that it was correctly targeted to the LDs.

Line 464 - "This also fits with what was observed in N. benthamiana [11] and properly arise from a change in the kinetics for the PTS to yield lower turnover numbers when this attached to the LD's": Consider rewording this sentence, Current wording is confusing and/or grammatically incorrect(?)

Answer: All the corrections have been incorporated and the text has been clarified.

---

## [Editor Report · Decision Letter 2]

25 Nov 2020

Connecting moss lipid droplets to patchoulol biosynthesis

PONE-D-20-29050R2

Dear Dr. Simonsen,

We’re pleased to inform you that your manuscript has been judged scientifically suitable for publication and will be formally accepted for publication once it meets all outstanding technical requirements.

Kind regards,

Björn Hamberger

Academic Editor

PLOS ONE

Additional Editor Comments (optional):

I am excited to see this improved manuscript accepted and look forward seeing this solid study published.
---

## [Editor Report · Acceptance letter]

27 Nov 2020

PONE-D-20-29050R2 

Connecting moss lipid droplets to patchoulol biosynthesis 

Dear Dr. Simonsen:

I'm pleased to inform you that your manuscript has been deemed suitable for publication in PLOS ONE. Congratulations! Your manuscript is now with our production department. 

Kind regards, 

on behalf of

Dr. Björn Hamberger 

Academic Editor

PLOS ONE